# Dating Violence among High-Risk Young Women: A Systematic Review Using Quantitative and Qualitative Methods

**DOI:** 10.3390/bs6010007

**Published:** 2016-01-29

**Authors:** Lauren E. Joly, Jennifer Connolly

**Affiliations:** Department of Psychology, York University, 5022 TEL, 4700 Keele St. Toronto, ON M3J1P3, Canada; connolly@yorku.ca

**Keywords:** high-risk, dating violence, young women

## Abstract

Our systematic review identified 21 quantitative articles and eight qualitative articles addressing dating violence among high risk young women. The groups of high-risk young women in this review include street-involved, justice-involved, pregnant or parenting, involved with Child Protective Services, and youth diagnosed with a mental health issue. Our meta-analysis of the quantitative articles indicated that 34% (CI = 0.24–0.45) of high-risk young women report that they have been victims of physical dating violence and 45% (CI = 0.31–0.61) of these young women report perpetrating physical dating violence. Significant moderator variables included questionnaire and timeframe. Meta-synthesis of the qualitative studies revealed that high-risk young women report perpetrating dating violence to gain power and respect, whereas women report becoming victims of dating violence due to increased vulnerability.

## 1. Introduction

It is widely known that a young woman’s personal background and life experiences impact her chances for both positive and negative outcomes. Key “background factors” that can lead to negative outcomes include living in unstable housing, family violence, and mental health issues [1]. Due to these background factors, certain young women are considered to be “high-risk” and are, thus, at increased odds for a range of poor outcomes, such as low educational achievement, interpersonal difficulties, and maladjusted coping [2]. There is a substantial body of literature on the poor outcomes high-risk young women can experience. Recently, researchers have begun to examine the occurrence of dating violence in the lives of these young women (ex. [3]). Dating violence includes acts of emotional, physical, or sexual violence in a romantic or sexual relationship [4]. Dating violence is a key issue for young women as it is associated with many negative and long-lasting consequences [5].

### 1.1. High-Risk Status

“High-risk” refers to youth, ages 12–25 years old, who are more likely to experience interpersonal and intrapersonal distress, such as dating violence, as a result of adverse environmental circumstances [6]. At present, the literature is silent as to which groups of high-risk young women might experience elevated rates of dating violence. However, background-risk factors related to dating violence have been consolidated in two key literature reviews. These reviews identified the following risk factors: familial violence, unstable living conditions, involvement in criminal activity, and mental health issues [4,7]. Based on these risk factors five groups of young women were identified in this review, through a preliminary search of the literature, as potentially at high-risk for dating violence. These are: street-involved, justice-involved, pregnant or parenting, involved with Child Protective Services (CPS), and diagnosed with a mental health issue (e.g., conduct disorder, depression, suicidal ideation).

### 1.2. Dating Violence

Dating violence among youth refers to a range of behaviours aimed at harming a partner. Three main types of dating violence are highlighted in the literature. The first is emotional (or psychological) violence, which includes actions such as damaging a partner’s belongings and insulting a partner in front of others. The second is physical violence which involves slapping, scratching, shoving, choking, beating, and assault with a weapon. The third is sexual violence, such as forcing a partner to have sexual intercourse or engage in other sexual acts against the partner’s will [8].

The prevalence of dating violence among adolescent girls who are not considered high-risk is estimated at 15% to 20% for victimization and 20% to 30% for perpetration [9,10]. In addition to different prevalence rates, dating violence victimization and perpetration are qualitatively distinct experiences. According to a review by Vezina and Herbert, women who stay in a romantic relationship where they are victimized are more likely to report feeling stronger love for their partner, having more traditional attitudes about gender roles, and more justification for their partners’ violence compared to women who leave violent relationships. Additionally, women likely perpetrate violence against their partners in self-defense, and respond to their partner’s violence with more violence [7].

The past twenty years have seen an emerging body of literature on high-risk young women and dating violence; however, to date, no literature reviews have been conducted solely on this research. A brief survey of the high-risk literature reveals prevalence rates of dating violence among high-risk young women ranging from 12% to 68% for victimization and 34% to 67% for perpetration [11,12,13]. Given this variation, it is important to conduct a review of the literature in order to derive a more comprehensive and holistic analysis. Thus, a central goal for this review is to determine the prevalence rate of dating violence perpetration and victimization among groups of high-risk young women, and whether some of these groups are at greater risk than others.

### 1.3. Factors Moderating the Prevalence Rate of Dating Violence

Dating violence rates are highly variable in community samples with both demographic and methodological factors influencing its occurrence [4,7]. Four key variables, age, ethnicity, questionnaire, and timeframe, have been shown to impact the occurrence of dating violence in community samples and, thus, are likely to influence its prevalence in high-risk groups as well.

#### 1.3.1. Age

Dating violence perpetration and victimization have been found to increase with age. However, this pattern may not hold for all forms of dating violence. For example, moderate and severe physical and sexual violence perpetration have been shown to peak at around 17 years of age, and then decrease in young adulthood. Despite some variation in rate, intimate partner violence can continue across developmental stages, as it has been shown that adolescent girls who experience dating violence victimization are significantly more likely to continue as both victims and perpetrators of dating violence in young adulthood [14].

#### 1.3.2. Ethnic Minority Status

Ethnic minority group has been identified in several studies as a moderator for the prevalence rate of dating violence victimization and perpetration. Ethnic minority status refers to youth who report a different racial or cultural group from the majority population where they reside [15]. According to Capaldi *et al.*’s review article [4], victimization and perpetration rates are higher among African American youth, although these differences diminish after controlling for reduced socio-economic status and income. Similarly, in the Vezina and Herbert review article [7], some studies found that women with African American, Hispanic, and Asian-American backgrounds were at increased risk for dating violence victimization, while other studies identified those same groups to be at decreased risk. Thus, while results appear to be mixed in dating violence research with community samples it is, nonetheless, still important to examine this variable among high-risk young women, because the effects may be clearer among youth from high-risk *versus* community samples.

#### 1.3.3. Dating Violence Questionnaire

The measurement of dating violence varies among research studies, and this may impact the prevalence rate reported. There are two well-accepted multi-item questionnaires used in the dating violence literature. These include The Conflict Tactics Scale [16], which was later revised, and more recently the Conflict in Adolescent Relationships Inventory [10]. Alternatively, researchers may create their own dating violence indices to meet their particular research needs, and ask individually-targeted questions related to specific aspects of dating violence [17]. These two approaches differ in the number of items used, with dating violence scales containing multiple questions and researcher-generated indices typically asking the youth one question regarding whether or not they have experienced dating violence [18].

#### 1.3.4. Timeframe

Additionally, some researchers ask youth to report on the violence they have experienced within a specific timeframe, while others require youth to report on their entire life’s experiences. For example, in Wolfe *et al.* [10], participants were surveyed regarding behavior within the past six months, whereas participants in Howard, Debnam, Wang, and Gilchrest [19], reported on ever experiencing a particular behavior. A meta-analysis of dating violence among lesbians found that prevalence rates for both victimization and perpetration were higher for articles that reported dating violence across the lifespan *versus* the past year [20]. While LGBT couples are not considered a high-risk group for dating violence in this current study, it is likely that articles that report prevalence rates across the lifespan will also have greater rates of dating violence for high-risk young women.

### 1.4. Qualitative Research on Dating Violence

In addition to examining prevalence rates and moderators, researchers have explored the qualitative experiences of dating violence. In these qualitative studies, youth from community samples define dating violence as actions including yelling, name-calling, ignoring, shoving, hitting, violence with a weapon, and unwanted sexual advances or activity. Dating violence experienced by the youth was described as the perpetrator attempting to control the victim through threats, violence, and strict boundaries. It also involved one or both parties feeling judged or disrespected by their partner. Dating violence was found to either consist of one partner as the perpetrator and the other as the victim, or both partners perpetrating acts of violence against each other [21,22]. Women in the articles who were victims of dating violence reported feeling reluctant to seek help due to fears of retaliation from their partners, and judgement from friends and family [23]. Given the relevance of this information in understanding the process of dating violence, another central focus of this project is to synthesize the qualitative experiences of high-risk young women in violent relationships. Issues of control, disrespect, judgement and powerlessness will likely be even more pronounced among high-risk young women.

### 1.5. Analyzing Quantitative and Qualitative Research: A Mixed Methods Approach

The main goal of this study is to compile the literature on the identified groups of young women at high-risk for dating violence in order to determine the extent and nature of their experiences with dating violence. In this systematic review of the literature quantitative articles will be analyzed through meta-analysis and qualitative articles will be analyzed through meta-synthesis. The meta-analysis will result in overall or “global” prevalence rates which will demonstrate the proportion of high-risk youth who have experienced dating violence victimization and perpetration. The meta-synthesis will explore qualitative articles, to assess motivations and personal factors related to dating violence. The intention of the qualitative analysis is to allow the themes to develop organically from the articles themselves. Conducting both quantitative and qualitative analyses will provide a more complete understanding of this body of literature.

#### 1.5.1. Research Questions: Meta-Analysis

(1)What proportion of high-risk young women have perpetrated dating violence and what proportion have been victims of dating violence?(2)Does the type of high-risk group moderate the proportion of young women involved in dating violence?(3)Do the following moderator variables: mean age, percentage of ethnic minorities, questionnaire type, and timeframe, affect the proportion of young women who have experienced dating violence?

#### 1.5.2. Research Questions: Meta-Synthesis

(1)What are the key themes highlighted in the qualitative articles related to high-risk young women’s experiences of dating violence?(2)How do these young women define dating violence?(3)What are high-risk young women’s motivations for dating violence?(4)Are there personal or social factors that influence dating violence?

## 2. Method

### 2.1. Systematic Literature Search

The identification of articles was completed by searching the following social science databases: PsycINFO, ERIC, and Social Sciences Abstracts, using the key words *at-risk or high-risk and youth or adoles* or teen* and dat* or intimate and violence or aggression.* Rather than include specific high-risk groups in the search terms, more general terms were used in order to ensure that groups were not missed and to determine if unexpected groups were identified. The search resulted in 976 articles. The following inclusion/exclusion criteria were used to select articles based on the information in the abstract (first pass): a quantitative or qualitative research methodology, a sample of youth aged 12–25 years, and publication in the English-language. Additionally, the article specifically addressed dating violence, and the sample included female youth from one or more of the high-risk groups: street-involved, justice-involved, pregnant or parenting, involved with Child Protective Services, and diagnosed with a mental health issue. A second screening (second pass) was conducted by a thorough reading of each article to ensure that it met inclusion criteria. See Figure 1 for the number of articles discarded at each pass, and the percent of articles discarded based on exclusion criteria.

### 2.2. Meta-Analysis

Quantitative articles with female youth who have experienced dating violence were analyzed through meta-analysis. Two effect sizes were calculated: one for the overall proportion of high-risk female youth who have experienced dating violence victimization, and one for those who have engaged in perpetration. Additionally, meta-regressions were conducted to explore the moderating effects of group type, ethnic minority, mean age, questionnaire, and time frame.

**Figure 1 behavsci-06-00007-f001:**
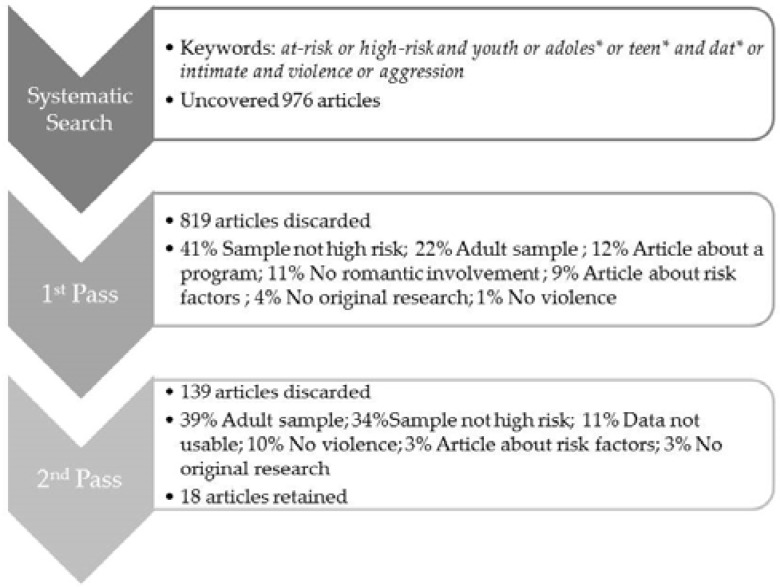
Systematic Literature Search Process. *Notes: some articles may have been discarded based on multiple criterion, however for the purposes of calculating percentages, each article was only categorized by one criterion.

The following information was coded for each study: (1) total number of participants; (2) high-risk group; (3) mean age of the sample; (4) percent of ethnic minority participants in the sample; (5) type of questionnaire used to measure dating violence (1–2 item *versus* multi-item); and (6) timeframe during which the violence may have occurred (2–12 months *versus* lifetime). Following Card [24], articles were re-coded six months later by the primary researcher and intra-rater reliability was 98%. Contradictions were addressed by consulting the coding manual to determine where and why the mistake occurred [24,25].

### 2.3. Meta-Synthesis

The qualitative articles were analyzed to identify key themes. Meta-synthesis strives to maintain the integrity of the results within the original studies, while integrating them across studies to highlight common themes. The meta-synthesis was conducted using the meta-ethnographic principles of Noblit and Hare [26] and Malpass *et al.* [27]. These authors outline three steps following the systematic literature search, which we followed: (1) noting original quotes and themes from the selected studies. Quotes that were a clear example of the key themes of the article were selected; (2) identifying common key themes among the studies and determining how the studies are related to one another. This involved assessing the themes from each study to determine if they agree, disagree, or build upon each other to form a continuous argument; and (3) synthesizing the identified relationships. This involved determining if new, overarching themes could encompass the themes identified in the original articles.

## 3. Results

### 3.1. Selected Articles

Twenty-one of the articles contained quantitative data reporting the proportion of young women involved with physical dating violence (see Table 1), and eight of the articles contained qualitative data (see Table 2). None of the articles reported both quantitative and qualitative data. The high-risk groups found in the articles street-involved, justice-involved, pregnant or parenting, involved with Child Protective Services, and diagnosed with a mental health issue. One qualitative article [28] discussed violence within a lesbian relationship between women who were also involved in the juvenile justice system.

**Table 1 behavsci-06-00007-t001:** Summary of Meta-Analysis Sample Characteristics [3,11,12,13,29,30,31,32,33,34,35,36,37,38,39,40,41,42,43,44,45].

Authors	Number (Mean Age)	Dating Violence Question-Naire	Timeframe	High-Risk Type	Ethnic Minority (%)	Victims of Dating Violence (%)	Perpetrators of Dating Violence (%)
Buttar, Clements-Noel, Haas, and Reese (2013)	305 (16)	Single item	Lifetime	Juvenile Justice	58	33.5	NA
Chase, Treboux, and O’leary (2002)	31 (16)	CTS 18 item	Most recent partner	Delinquent	31	NA	34
Collin-Ve’zina, Herbert, Manseau, Blais, and Fernet (2006)	220 (16)	CTS-R	Lifetime	CPS	NA	60	NA
Gavin, Lindhorst, and Lohr (2011)	173 (17)	CTS 7 item	Lifetime	Pregnant/Parenting	47	68	NA
Hovespian, Blais, Manseau, Otis, and Girard (2010)	328 (16)	CTS-R	Lifetime	CPS	29	68	NA
Kelly, *et al.* (2009)	590 (15)	VDR	Lifetime	Juvenile Justice	89	51	NA
Lindhorst, Beadnell, Jackson, Fieland, and Lee (2009)	240 (22)	CTS	Past 6 months	Pregnant/Parenting	47	48	NA
Lipsky, Holt, Easterling, and Critchlow (2004)	682 (NA)	Single item	Past 9 months	Pregnant/Parenting	NA	12	NA
Milan, *et al.* (2005)	163 (17)	CTS-2 12 item	Past 3–12 months	Pregnant/Parenting	91	34	43
Moretti, Obsuth, Odgers, and Reebye (2006)	63 (15)	CTS 6 item	Past 3–12 months	CPS	33	NA	34
Mylant and Mann (2008)	49 (18)	AAS	Lifetime	Pregnant/parenting	100	61	NA
Raneri, Leslie, Wiemann, and Constance (2007)	581 (17)	Single item	Past 3 months	Pregnant/Parenting	70	20	NA
Rizzo, Esposito-Smythers, Spirito, and Thompson (2010)	118 (15)	CADRI 35 item	Past 3–12 months	In-patient	19	28	NA
Siefford (1997)	162 (18)	Single item	Lifetime	Juvenile Justice	77	15	NA
Slesnick, Erdem, Collins, Patton, and Buettner (2010)	99 (19)	BRFSS	Lifetime	Street-Involved	81	36	NA
Wekerle, *et al.* (2001)	107 (16)	CIRQ- SF	Past year	CPS	26	19	67
Wekerle, *et al.* (2009)	180 (16)	CADRI 80 item	Past year	CPS	71	63	67
Wenzel, D’Amico, Barnes and Gilbert (2009)	27 (21)	Single item	Past 3–12 months	Street-Involved	93	22	NA
Wiemann, *et al.* (2000)	724 (15)	AAS	Past year	Pregnant/Parenting	70	12	NA
Wolfe, *et al.* (2003)	32 (15)	CADRI	Past 2 months	CPS	15	NA	41
Yang, *et al.* (2006)	111 (17)	AAS	Past 3–12 months	Pregnant/Parenting	100	12	NA

*Notes: CTS = Conflict Tatics Scale; VDR = Victimization in Dating Relationships Instrument; AAS = Abuse Assessment Screening form; CADRI = Conflict in Adolescent Dating Relationships Inventory; BRFSS = Behavioral Risk Factor Surveillance System; CIRQ-SF = Conflict in Relationships Questionnaire-short form; CPS = Child Protective Services.

**Table 2 behavsci-06-00007-t002:** Summary of Meta-Synthesis Sample Characteristics [28,46,47,48,49,50,51,52].

Authors	Location	Sample N	Age Range	High-Risk Group
Ashley (1998)	United States	1	18 years	Juvenile justice
Bourgois, Prince, and Moss (2004)	United States	2	20 years	Street-involved
Brown, Brady, Letherby (2011)	United Kingdom	9	16–23 years	Teenage mothers
Kidd and Kral (2002)	Canada	29	17–24 years	Street-involved
Miller, Levinson, Herrera, Kurek, Stofflet, and Marin (2011)	United States	17	19–23 years	Maltreated (CPS)
Saewyc (1999)	United States	8	17–19 years	Street-involved and pregnant
Schaffner (2007)	United States	1	15 years	Juvenile justice
Simkins and Katz (2002)	United States	26	-	Juvenile Justice

The quantitative articles reported proportions of young women experiencing psychological, physical, and/or sexual dating violence. Thirteen of the 21 quantitative articles reported separate proportions for each type of dating violence, 10 articles reported proportions for combined types of dating violence, and two articles reported both combined and separate proportions (see Table 3). To address this heterogeneity we selected an effect size that was consistent across articles to ensure that the global effect size was an accurate representation of the construct being measured. By selecting a common measure it was possible to retain all of the articles in the analysis. This is important as a large sample of articles enhances the generalizability of the results [25].

**Table 3 behavsci-06-00007-t003:** Quantitative Articles Reporting a Prevalence Rate for “Types” of Dating Violence [3,11,12,13,29,30,31,32,33,34,35,36,37,38,39,40,41,42,43,44,45].

Authors	Physical Abuse	Sexual Abuse	Emotional Abuse	Abuse Types Combined (*i.e.*, Physical and Sexual)
Buttar, Clements-Noel, Haas, and Reese (2013)				V (phys. and sex.)
Chase, Treboux, and O’leary (2002)				P (phys. and emo.)
Collin-Ve´zina, Herbert, Manseau, Blais, and Fernet (2006)	V	V	V	
Gavin, Lindhorst, and Lohr (2011)				V (phys. and threats)
Hovespian, Blais, Manseau, Otis, and Girard (2010)	V	V	V	
Kelly, *et al.* (2009)		V		V (phys. and sex.)
Lindhorst, Beadnell, Jackson, Fieland, and Lee (2009)				V (phys. and threats)
Lipsky, Holt, Easterling, and Critchlow (2004)				V (phys. and emo.)
Milan, *et al.* (2005)	V			
	P			
Moretti, Obsuth, Odgers, and Reebye (2006)	P			
Mylant and Mann (2008)				V (phys. and sex.)
Raneri, Leslie, Wiemann, and Constance (2007)	V	V		
Rizzo, Esposito-Smythers, Spirito, and Thompson (2010)			V	V (phys. and sex)
Siefford (1997)	V			
Slesnick, Erdem, Collins, Patton, and Buettner (2010)	V	V	V	
Wekerle, *et al.* (2001)	V	V	V	
	P	P	P	
Wekerle, *et al.* (2009)				V (phys., sex., emo.)
				P phys., sex., emo.)
Wenzel, D’Amico, Barnes and Gilbert (2009)	V		V	
Wiemann, *et al.* (2000)	V			
Wolfe, *et al.* (2003)	P			
Yang, *et al.* (2006)	V			

* Notes: V = reports a proportion for victimization; P = reports a proportion for perpetration.

To determine the common metric we turned to the literature for guidance. Empirical research indicates that young women are more commonly involved in physical dating violence than sexual dating violence [53]. Consistent with this, the majority of the articles in our study reported victimization and perpetration rates for physical dating violence, thus we calculated the global prevalence rates using the proportions described as containing “at least” physical dating violence, (e.g., a proportion could be comprised of physical victimization alone, or physical and sexual victimization). This ensured that the effect sizes selected from each article were similar in that they all contained physical dating violence, but also permitted the largest sample size possible. No articles were excluded based on this decision, as all articles that met the other inclusion criteria contained proportions for physical violence (see Table 3). Of the articles with relevant quantitative data, 18 reported victimization rates, and six reported perpetration rates. The victimization rates ranged from 12% to 68%, and the perpetration rates ranged from 22% to 68%.

### 3.2. Meta-Analysis

#### 3.2.1. Statistical Analysis

Individual global effect sizes were calculated for the rate of high-risk young women reporting victimization of physical dating violence, and the rate of high-risk young women reporting perpetration of physical dating violence. The proportion of young women reporting victimization and perpetration was calculated for each article, then transformed into a logit, and finally weighted by the standard error for analysis. The global effect sizes were calculated using a random effects model; effect size and confidence intervals were then back-transformed from a logit to the original metric, for the purposes of interpretation [24].

#### 3.2.2. Global Effect Sizes

Calculation of the global effect size for victimization, *p* (proportion) = 0.34 (CI = 0.24–0.45), indicated that 34% of high-risk young women have been victims of physical violence by a romantic partner. Follow-up analysis demonstrated that the effect size was not homogeneous, (*Q_w_* = 5177.20, *p* < 0.001), highlighting that the variance among effect sizes was greater than would be expected due to sampling error alone. The global effect size for perpetration, *p* (proportion) = 0.45 (CI = 0.31–0.61), indicated that 45% of high-risk young women have perpetrated physical dating violence. Again, this effect size was not homogenous (*Q_w_* = 311.96, ρ < 0.001), thus supporting the analysis of moderators to assess the sources of variance [24].

#### 3.2.3. Moderating Effects of High-Risk Group

Meta-analysis regressions were then estimated using multilevel modeling to determine if the type of high-risk group affected the proportion of young women who perpetrated or were victims of dating violence. The data for the regression analyses was also transformed into a logit before analysis, and the regression slopes and standard errors were then back transformed into a factor (exponentiation) of the original metric after the analysis for ease of interpretation. Results were non-significant (see Table 4) indicating that the high-risk group within which a young woman is categorized does not impact her odds of experiencing dating violence.

**Table 4 behavsci-06-00007-t004:** Meta-regression Analysis.

Violence	Variable	β	SE (*B*)	*p*-Value
*Victimization*				
	High-Risk Group: juvenile justice vs. street-involved	0.48	0.67	0.89
	High-Risk Group: juvenile justice vs. pregnant/parenting	0.48	0.72	0.91
	High-Risk Group: juvenile justice vs. CPS	0.70	0.69	0.29
	High-Risk Group: juvenile justice vs. inpatient	0.46	0.77	0.90
	% Ethnic Minority	0.50	0.50	0.51
	Mean Age of Sample	0.50	0.53	0.90
	Questionnaire: 1–2 questions vs. multi-item questionnaire	0.62	0.60	0.0013
	Time Frame: 2–12 months vs. lifetime	0.74	0.60	0.01
*Perpetration*				
	High-Risk Group: pregnant/parenting vs. violent	0.97	0.77	0.38
	High-Risk Group: pregnant/parenting vs. cps	0.85	0.72	0.90
	% Ethnic Minority	0.51	0.50	0.49
	Mean Age of Sample	0.55	0.62	0.69

#### 3.2.4. Moderating Effects of Age, Ethnicity, Questionnaire, and Timeframe

Meta-analysis regressions were also estimated using multilevel modeling for the mean age of the youth in the sample and percent of ethnic minority youth in the sample, for both victimization and perpetration (see Table 4). The variables, questionnaire type (1–2 item *vs*. multi-item) and time frame (2–12 months *vs*. lifetime) could only be analysed for victimization as all articles reporting prevalence rates for perpetration contained data from multi-item questionnaires with a restricted timeframe (see Table 4).

The regression analyses indicated that “questionnaire type” significantly moderated the proportion of high-risk young women who report dating violence victimization, such that the odds of reporting physical dating violence victimization were higher with multi-item questionnaires compared to questionnaires with only 1–2 items, b = 0.62, *p* < 0.01. The time frame or reference period used in the study also significantly moderated the proportion of high-risk young women who report dating violence, such that the odds of reporting physical dating violence victimization were higher when the questionnaire addresses the youths’ experiences over their entire life *versus* experiences within the past year, b = 0.74, *p* < 0.05.

### 3.3. Meta-Synthesis

Eight articles were identified for inclusion in the meta-synthesis. Based on data from these articles three overarching themes were identified. These included: perceptions of dating violence and romantic relationships; motivations for dating violence; and factors influencing dating violence. These themes are discussed in the following sections.

#### 3.3.1. Perceptions of Dating Violence

Five of the eight articles addressed high-risk young women’s perceptions of actions that constitute dating violence and what dating violence means in the context of their romantic relationships. All of these articles described acts of victimization. The women in these five articles described their partners’ actions as including slapping, choking, pushing, beating, swearing, controlling, threats, isolation, intimidation, put-downs, slamming doors, and forced sex [48,49,51,52]. Four of the five articles also described dating violence perpetrated by young women, with actions including, stabbing, stalking, screaming, hitting, kicking, and pointing a gun [47,48,49,53]. Much of the dating violence perpetrated by these women resulted from perceived aggressive actions by their partner, *i.e.*, “when her boyfriend raised his hand to her, she showed him her bicycle chain and said she would kill him if he did it again” [48] (p. 260).

For the majority of the women in the five articles reporting their perceptions, dating violence was viewed as an expected part of the romantic relationship. Some of the women in the articles tolerated the abuse by minimizing their partners’ actions. One woman, in the article by Saewyc, denied that her experiences were dating violence due to lack of severity: “it wasn’t like as bad as somebody that’s abusive[…] he barely slapped me” [52] (p. 114). While another woman in Miller *et al.* trivialized her experiences due to lower frequency: “I would always be like oh well he doesn’t hit me all the time, I thought it was normal” [51] (p. 78). Other women in the articles took this a step further, and viewed abusive actions not only as an accepted part of the romantic relationship, but as a sign of their partners’ love. Forced sex and physical abuse from their partners were seen as an indication of caring, love, commitment, protection, and even ownership [48,52]. One woman proudly explained, “the harder he hits you, the more he loves you” ([48] p. 260). It is important to note however, that a minority of the women viewed aggression from their partners as a negative and unacceptable behavior, stating “I felt like he forced me. For me, that was a rape” [49]; [51] (p. 79). Thus, while some women in the studies perceived dating violence as a negative aspect of their romantic relationships, others understood it as an accepted or even loving part of their relationships.

#### 3.3.2. Motivations for Dating Violence

Three explanations for women’s involvement in dating violence were identified in the articles. These motivations include feelings of vulnerability, feelings of disrespect or judgment, and inequality in the relationship. All of these themes were found to play a role in both perpetration and victimization experiences.

##### Vulnerability

Seven of the eight articles identified feeling vulnerable as playing a key role in both perpetration and victimization for high-risk young women. Women who perpetrated dating violence often did so in order to feel powerful and gain some measure of control over their circumstances. One woman stated, “I felt good after. When I was hitting him I had so much power” [47] (p. 108). Women in all of the seven articles reported acting violently due to feelings of helplessness, vulnerability, and difficulty accepting their partners’ decision to end the relationship. For example, as described in one article, “Tanya was a 14-year old girl. When her 19-year old boyfriend told her that their relationship was over, Tanya walked home, got a kitchen knife, and returned to stab him in the jugular vein” [53] (p. 1487). The women disliked feeling vulnerable, and so sought power through engaging in aggressive, masculine behavior [47,53]. Women in the articles also reported engaging in aggressive behavior when their partner became aggressive with them [48,49].

Women in the seven articles reported experiencing dating violence from their partners due to being vulnerable and lacking agency in a situation. Teen mothers remained with violent partners for fear of losing their children, and street-involved young women remained in aggressive relationships for protection from further violence on the street [48,49]. For example, according to one woman in Burgeois *et al.* [48] (p. 257), “girls are afraid that if they don’t […] kick it with a guy then […] the next guy is going to come along and like […] you know hassle them”. The role vulnerability plays is particularly evident with regards to forced sex. Women in the articles reported feeling expected to have sex and unable to refuse partners’ advances [53]. For example, one woman in Saewyc [52] (p. 81) stated, “I mean cuz I said no a numerous amount of time, and they always end up taking it”. Women also indicated that their partners trapped them in the relationship through pregnancy, either by enticing the women to become intoxicated so they failed to think about contraception, or by refusing to wear a condom [49,51].

##### Disrespect and Judgement

Five of the eight articles highlighted the role that feelings of being disrespected and judged played in high-risk young women’s experience of dating violence, both for victimization and perpetration. The women in the articles reported being victims of violence because their partners felt disrespected. For example, according to one woman, “he found out that I had sex with somebody else and he slapped me” [52] (p. 87). Additionally, women in the articles reported remaining in violent relationships because they felt judged by other people, and so needed to defend their choice of partner. For example, a woman in one article explained, “the more and more people told me [to leave him], the more and more I was determined to stay with him to prove them wrong” [49] (p. 368).

Feeling disrespected also played a large role in the young women’s motivations to perpetrate dating violence. One woman in Ashley [47] (p. 114), discussed feeling triggered by perceptions of others’ disrespectful judgements, stating “I feel everyone is always talking about me, and that provokes me”. The women may even retaliate if they feel disrespected by their partners. Some women threatened to kill their partners because of cheating, abuse, or stealing from them. They did not report remorse for their actions, as one woman reported, “he brought this on himself,” with regards to perpetrating violence against her partner [47] (p. 113); [48,51].

##### Inequality in the Relationship 

Three of the eight articles identified inequality in the relationship as contributing to both perpetration and victimization of dating violence. Some women who perpetrated dating violence had difficulty seeing others’ worth, and instead viewed partners as “frustrating objects to be pushed aside or destroyed” [47] (p. 112). Unsurprisingly, some women who were victims of dating violence had the opposite viewpoint, and considered their partners’ desires to be more worthwhile than their own. Women reported having sex with their partners, despite not wanting to, in order to make their partners happy, and viewed sex as being for their partners’ pleasure *versus* their own [52]. A key factor which led to this inequality was a large gap between partners’ ages. Women in relationships with partners much older than themselves (ex. 16 years old *versus* 33 years old) [49], reported partners treating the women like children or less knowledgeable. As one woman in Brown *et al.* [49] (p. 368) explained, “[he was] abusive in the way he treated me […] because he thought he was older and he thought I was a child. […] he used to twist my emotions and make me feel guilty sometimes as if I had done something wrong […] until I was the one saying sorry”. The women felt that they had to accept the abuse because their partner was older and, therefore, knew best.

#### 3.3.3. Factors Influencing Dating Violence

Two key factors that are related to women’s involvement in dating violence were identified in the articles. These include personal factors and social-relationship factors. Both were found to play a role in women’s perpetration and victimization experiences.

##### Personal Factors

Four of the eight articles highlighted the role personal factors play in a young woman’s experience of dating violence. In addition to identification as part of a high-risk group, such as being a teenage mother or involved in the juvenile justice system, these young women reported other personal difficulties. These included: truancy, experiencing hallucinations and delusions, irrational thinking, drug use, suicidal ideation, self-harm, poor peer relationships, aggression towards peers and animals, and low self-esteem. These issues may make women more vulnerable to experiencing or participating in violent relationships. It was also noted that post-traumatic stress disorder may lead to aggression if the woman experiences flashbacks to earlier trauma, as this may lead to confusion between past and present partners [47,51,52,53].

##### Influences for Dating Violence: Family and Peer Violence

Seven of the eight articles highlighted common social-relationships experienced both by women who perpetrated dating violence and those who were victims. The women stressed that they learned how to treat others and how to let others treat them from their parents. They indicated that their parents did not model appropriate coping skills, anger management, or self-validation [47,52,53]. The women also felt that their parents failed to model healthy relationship skills, as one woman in Miller *et al.* [51] (p. 80), stated “I think what I saw with my parents is affecting my relationship a lot”. The cycle of violence is clearly prevalent in the experiences of high-risk young women, as one woman in Schaffner [28] (p. 1235), explained “I don’t know how else to say it. I get drunk and kick my girlfriend’s ass just like my dad gets drunk and kicks my mom’s”.

The women also discussed negative experiences with peers in their neighborhoods. They reported interpersonal difficulties, gang involvement, and witnessing violence between friends. Due to the high level of violence around them, the women in the articles appeared to have become desensitized to aggression. Thus they came to view aggressive acts against themselves or perpetrated by themselves towards others as lacking seriousness. This denial made them less likely to seek help for violence in their dating relationships. [47,51,53]. The women also noted the impact of their living situation on their dating relationships. Teenage mothers living with their partner’s families reported experiencing violence not only from their partners, but also from their partners’ parents: “I was pregnant the last time I got physical abuse off his mom. She hit me […]” [49] (p. 369).

## 4. Discussion

The goal of this review was to summarize the literature on high-risk young women and dating violence using meta-analytic and meta-synthetic methods. Two critical findings were obtained. First the prevalence rates for both dating violence perpetration and victimization were high. Second these young women reported numerous vulnerabilities and relational influences that contribute to their experiences of dating violence.

Overall, more than one third of high-risk young women reported being victims of dating violence and almost half reported perpetrating dating violence. While it may appear surprising that the perpetration rates are greater than victimization rates, this pattern is consistent with young women in the general population [9,10]. The perpetration rates may be greater than the victimization rates for several reasons. Firstly, this is potentially the result of the sample of articles identified for analysis. The analysis for victimization was conducted on 18 articles with a range of 12% to 68% for victimization rates, whereas the perpetration analysis was conducted on six articles with a range of 34% to 67% for perpetration rates. Given that the perpetration rate was calculated using only six studies, we have less confidence in the reliability of this rate, compared to the victimization rate which was calculated with three times more studies. Additionally, the victimization articles with smaller proportions had large sample sizes (*i.e.*, 682 participants) [12], and so had a large impact on the overall mean rate. A second explanation for the lower victimization rate may be found by considering the qualitative articles in this sample. Women who described dating violence perpetration clearly acknowledged their actions, whereas women who described victimization experiences tended to minimize their partners’ actions. Therefore, it is possible that in the quantitative articles women were similarly underreporting their victimization experiences, while more accurately reporting their perpetration experiences.

It is important to note that the prevalence rates for victimization are impacted by how the women’s experiences are measured. Studies that use multi-item questionnaires are more likely to report higher rates than studies with only one or two questions on dating violence. This is likely because multi-item questionnaires ask the women about numerous specific experiences, *versus* simply asking whether or not they have experienced dating violence. These multi-item questionnaires are then more likely to elicit responses from women who, as discussed above, are willing to say they have been slapped, for example, but not that they have been abused. Additionally studies that permit the youth to answer based on their entire life are more likely to have higher victimization rates compared to studies that limit the youth to only reporting their experiences in the past month or year. This provides key implications for research regarding the importance of selecting accurate and valid measures of dating violence. These two moderator variables could not be assessed for perpetration because all of the articles fell into the same categories on these two variables. However, this may also explain why perpetration rates are higher than victimization rates, as all of the articles exploring perpetration used multi-item questionnaires.

Perpetration and victimization rates were not significantly altered by the different types of high-risk group. It is likely that group membership is not a significant moderator because many of the women fall into multiple high-risk categories, with teenage mothers also reporting living on the street or using drugs, for example. These non-significant findings indicate that it makes conceptual sense for this project to have combined these articles on seemingly different groups of young women. Perpetration and victimization rates were also not significantly moderated by mean age or percent of the sample identifying as an ethnic minority. The range of mean ages for the articles in the sample is relatively small (15–21 years old), which may have contributed to the non-significant findings. Additionally, given that previous literature has reported inconsistent findings on the role of ethnicity in dating violence, non-significant findings were not entirely surprising. More research is likely needed to better understand the role ethnicity plays in dating violence, if any [4,7].

Qualitative analysis provided considerable insight as to why dating violence rates were so high. High-risk young women reported intense emotions surrounding their involvement in both victimization and perpetration. Dating violence perpetration and victimization involved similar acts, such as stabbing, hitting, threatening, choking, pushing, intimidation, and rape, and the themes related to engaging in perpetration and being victimized were also similar.

Women reported perpetrating dating violence for many reasons: to feel in control and less vulnerable; because they felt disrespected or judged; or because they had difficulty viewing their partners as individuals of equal worth. Women also reported victimization because their partners felt disrespected and needed to restore justice, and because of inequality in the relationship, with older males dominating younger women [28]. Vulnerability also played a key role in women’s experiences of victimization. Many of the women reported staying with aggressive partners to avoid violence from others or because they have children who required protection. The theme of vulnerability was especially prevalent in articles addressing forced sex, as many women tried to refuse sexual advances, but reported feeling unable to voice their opinions on sex in the relationship with their partners.

Other negative influences experienced by high-risk young women are also likely to impact dating violence. The women discussed a number of personal factors, including mental health issues and difficulties with interpersonal relationships. The women also highlighted a number of social-relationship factors, including exposure to marital violence, poor parental modeling, low parental monitoring, aggressive peers, and living in violent neighborhoods. The articles clearly demonstrated interactions between internal traits and the external environment. Many of these factors are similar to those found in community samples [4]. However, these influences act as compounding factors, in that women who experience these other factors, in addition to belonging to one of our identified high-risk groups, may be at cumulative risk for dating violence.

### 4.1. Limitations

Despite the strengths of this systematic review, there are some limitations to the findings. One key limitation in conducting this meta-analysis was the varying methods the articles used for reporting proportions of dating violence. This limited the quantative analysis to focusing mainly on physical dating violence, instead of including separate analyses for sexual, emotional and physical violence. Additionally a much smaller number of articles reported perpetration compared to victimization, and the variables “timeframe” and “questionnaire” could not be analyzed for the articles reporting perpetration due to a lack of variability in the questionnaires used by the included articles. Another limitation is that the study does not examine dating violence among high-risk young men. In community samples, men’s motivations for perpetrating dating violence are different from women’s motivations. At present, there is limited research on men’s victimization experiences [29]. Thus, just as it is important to have an understanding of this issue among young women, it is necessary to explore whether prevalence rates, motivations, and influences are similar among high-risk young men.

### 4.2. Implications for Future Research

Future research goals may involve quantitatively exploring other moderator variables, such as living environment and mental health. It would also be beneficial to conduct more research to directly link motivations and life influences to dating violence prevalence among high-risk young women. Finally, only two of the qualitative articles in our sample [47,49] mentioned protective factors. It would be helpful to further explore what protects these young women from experiencing dating violence, or at least serves to mitigate their risk. Understanding which factors increase and decrease risk is key to addressing the highly prevalent issue of dating violence.

## 5. Conclusions

This systematic review consolidated the extant quantitative and qualitative literature on dating violence among high-risk young women. This mixed-methods approach provides a holistic understanding of the dating violence experiences of these women. By highlighting the elevated prevalence rates and the extensive vulnerabilities of these young women, our review provides significant insights into the relationship dangers they face as well as important directions for future research.

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
