# Peer review of "Dating Violence among High-Risk Young Women: A Systematic Review Using Quantitative and Qualitative Methods"

_behavsci, 2016, doi:10.3390/bs6010007_

Round 1
Reviewer 1 Report
This is an interesting paper that attempts to summarize the existing dating violence prevalence literature among high risk young women. While I think a review of this nature is needed, there are several areas of concern that lessen my enthusiasm for the manuscript in its present form.
1) My major concern is with the summary of estimates. I do not see how it is possible, or even accurate, to combine prevalence estimates from different time frames (e.g., ever, past year). This is like combining apples with oranges, and it is not a big surprise that the results differ by timeframe. I'm not sure what this is really adding to the literature.
2) Related to the above comment, the summary estimates presented in the abstract do not really mean anything unless there is a timeframe attached to them.
3) The introduction needs to be reorganized. The logical story of the article is not presented. For example, the authors should define what makes someone one high risk. For example, some supporting studies are among lesbian women, but it is not clear if the authors are saying that this target population is high risk. What does high risk mean? Then, move onto an understanding of what we know about dating violence among high-risk women (prevalence and risk factors). Give some data to support the risk factors and their association with dating violence. Then, discuss the consequences of dating violence, and be specific about what these consequences are. Be clear about the age range of the population that you are looking at up front.
4) The authors are using the term "moderators" incorrectly. The authors are looking at risk factors here, which are different. The authors are looking at whether the prevalence varies by certain factors. Be careful though about using the term "predictor" if data comes from cross-sectional studies.
5) For the tables, use the same high risk classification that you are using in the text. For example, I'm assuming that children's aid society is youth involved with cps - but that language should be used so the reader does not have to guess.
6) Minor comment - include shading every other line in tables. They are difficult to read as is.
7) On p. 18, the authors suggest that dating violence estimates among high risk women are higher than the general population. I'm not sure how the authors can come to this conclusion.
8) The authors should consider having an editor work with them to refine and polish the paper.
Author Response
Thank you for taking the time to review the paper, and provide helpful and considered comments. We have attached our responses to Reviewer 1's comments below.

Reviewer 2 Report
well written review on vulnerability of adolecent girls as an important factor in risk of dating violence victimization
Author Response
We thank the reviewer for taking the time to read our paper. We greatly appreciate your interest and support for our work.
Reviewer 3 Report
I found this paper to be very well-written and extremely interesting. I very rarely do this, but I really have no comments that I feel would help improve the paper. My personal opinion is that this manuscript is ready for publication as-is.
Author Response

(The authors gave the same response as above.)

Round 2
Reviewer 1 Report
The reviewer thanks the authors for their revision of this manuscript. It is now much improved and more clear in terms of its focus and results. There are a few minor comments that I think should be addressed before publication.
1) In the abstract, please include the confidence intervals for the prevalence estimates.
2) Under 1.2 Dating Violence, second paragraph, second sentence the following sentence is unclear: "In addition to different prevalence rates these relational roles are qualitatively distinct." What is meant by "these relational roles"? Are the authors referring to victimization vs perpetration?
3) Include a footnote for Tables where abbreviations are written out (e.g., CTS).
4) In 3.2.4, describe why measurement variables could only be examined for victimization. I know this is described in the discussion, but I think it needs to be placed here instead. It should also be listed as a limitation.
5) The additional limitation of only being to provide estimates for physical dating violence (and not emotional or sexual) should be stated.
6) There are still some minor edits I would make: 1) follow the same order when describing the list of high-risk groups throughout the paper 2) describe the high-risk groups exactly the same when describing them (e.g., sometimes described as CPS, sometimes described as child welfare) 3) similarly, describe the moderators the same way every time (e.g., in the abstract, the moderator variable is listed as measurement, but then later it is questionnaire type), 4) Retitle 3.2.4 to match listing of all moderators (e.g, age, percent minority) 5) match subtitles for 3.3.2 and 3.3.3 to how they are described under 3.3.
